# Assessment of temporomandibular disorders and their relationship with life quality and salivary biomarkers in patients with dentofacial deformities: A clinical observational study

Betina B. Crescente[1,2], Natalia V. Bisatto[1,2], Gabriel Rübensam[2], Guilherme G. Fritscher[3], Maria M. Campos[1,2]*

1 Programa de Pós-graduação em Odontologia, Escola de Ciências da Saúde e da Vida, Pontifícia Universidade Católica do Rio Grande do Sul, Porto Alegre, RS, Brazil, 2 Centro de Pesquisa em Toxicologia e Farmacologia, Escola de Ciências da Saúde e da Vida, Pontifícia Universidade Católica do Rio Grande do Sul, Porto Alegre, RS, Brazil, 3 Ambulatório de Cirurgia Oral, Escola de Ciências da Saúde e da Vida, Pontifícia Universidade Católica do Rio Grande do Sul, Porto Alegre, RS, Brazil

* maria.campos@pucrs.br, camposmmartha@yahoo.com

## Abstract

A close relationship between dentofacial deformities (DFD) and temporomandibular disorders (TMD) has been suggested, which might impact the quality of life (QoL) and psychological aspects. We evaluated the presence of TMD in DFD patients, correlating these findings with QoL and salivary levels of biochemical markers of pain and psychological disorders. The study enrolled 51 patients, which were distributed into three groups: (i) orthodontic, (ii) TMD, and (iii) DFD. TMD diagnosis was conducted according to Axis I and II of the *Diagnostic Criteria for Temporomandibular Disorders* (DC/TMD). QoL was evaluated by the *Oral Health Impact Profile* (OHIP-14). The salivary levels of interleukin-1β (IL-1β) were determined by ELISA, while glutamate and serotonin amounts were evaluated by mass spectroscopy. DFD individuals had a positive diagnosis for TMD, as indicated by the Axis I (DC/TMD). They exhibited poorer outcomes regarding pain, functional, and psychological dimensions, according to the Axis II DC-TMD. The QoL evaluation demonstrated poorer outcomes for DFD individuals, accompanied by greater IL-1β salivary contents. Notably, glutamate levels had a positive correlation with behavioral parameters in Axis II DC-TMD, with a mild relevance for serotonin. DFD patients display chronic myofascial pain featuring TMD, with altered psychological symptoms and poor QoL, encompassing changes in pain mediators. Data bring new evidence about the relevance of TMD in DFD patients, which likely impacts the QoL and the salivary levels of biochemical markers of functional, painful, and psychological disorders.

**Data Availability Statement:** All relevant data are within the paper and its Supporting Information files.

**Funding:** BBC - (001) Coordenação de Aperfeiçoamento de Pessoal de Nível Superior NVB - (001) Coordenação de Aperfeiçoamento de Pessoal de Nível Superior MMC - (304042/2018-8) Conselho nacional de Desenvolvimento Científico e Tecnológico The funders had no role in study design, data collection and analysis, decision to publish, or preparation of the manuscript.

**Competing interests:** The authors have declared that no competing interest exist.

## Introduction

Individuals carrying out dentofacial deformities (DFD) have mixed skeletal and dentition alterations. When the skeletal malocclusion is moderate to severe, a well-established treatment combines orthodontic therapy plus orthognathic surgery. This treatment aims to correct skeletal irregularities, restoring both masticatory function and esthetics, with psychological and social benefits [1]. The presence of DFD can be considered a risk factor for the development of temporomandibular disorders (TMD) [2]. Notably, TMD has a high prevalence in young and female individuals, ranging from 20% to 60% in children and adolescents, whom commonly search for orthognathic surgery [3].

The impact of orthognathic surgery on TMD alterations has been widely explored in the field of oral surgery [2], but this remains an issue of the current debate, that still requires further investigation. The management of TMD in general population might involve interdisciplinary approaches employing both pharmacological and non-pharmacological treatments [4]. For instance, a recent pilot study demonstrated that a combination of bilateral manual therapy and radial Extracorporeal Shock Wave Therapy (rESWT) resulted in a significant improvement of painful and functional parameters in individuals with TMD diagnosis. These therapies might well be useful for patients with dentofacial deformities and TMD symptoms, what remains to be further investigated [5].

TMD has been suggested to markedly affect the life quality of individuals, indicating that therapeutic interventions must address both physical and psychological conditions [6]. In addition to TMD, DFD has also been described to have negative impacts on the quality of life (QoL) of the affected individuals, which is allied with psychological conditions, such as depression. Of note, the presence of TMD in patients with DFD might contribute to impairment of QoL and mental health disorders, mainly due to the presence of painful symptoms [7]. Pertinently, depression and chronic pain can also affect the patient's perception of orthognathic surgery. In patients who present with a chronic pain state, the occurrence of depression likely increases the susceptibility to the development and perpetuation of pain [8]. Therefore, the binomial relationship between DFD and TMD, in the context of poor QoL and psychological symptoms, clearly indicates the complexity of managing DFD-affected individuals, beyond the orthodontic-surgical treatment. Adding intricacy to this scenario, individual variations in the ability to adapt to the mentioned comorbidities and the long-term treatment protocols are expected.

Recent studies have revealed that biological mechanisms, encompassing changes in molecular markers, are associated with poorer QoL levels in DFD patients. The interaction of cytokines and their receptors likely interferes with emotional functions. For instance, the interleukin-6 (IL-6) gene has been linked to low levels of QoL in DFD patients, being associated with the domains of emotional function, psychological discomfort, and social disability [9–11]. Besides that, salivary levels of glutamate showed a positive correlation with the functional dimension of the QoL in a group of DFD patients [12].

In the light of the current evidence, the present observational clinical study assessed the TMD outcomes of pre-surgery DFD patients, according to the Axis I and II of the *Diagnostic Criteria for Temporomandibular Disorders* (DC/TMD), correlating the findings with the levels of QoL and salivary biochemical markers of pain and psychological disorders. Our study shed new light on the status of patients with a DFD diagnosis, which might help to change the paradigms in the treatment of DFD individuals, adding up oral and psychological rehabilitation to orthognathic surgery.

## Materials and methods

### Study design, participants, and outcomes

This observational clinical study (pre-trial registration number 41678720.5.0000.5336) was performed in ethical agreement with the Helsinki Declaration (ninth version; [13]). All the procedures were approved by the Research Ethics Committee of the Pontificia Universidade Católica do Rio Grande do Sul (CEP-PUCRS; approval protocol 4.504.239), Brazil, in January 2021. Written informed consent was obtained from each patient, and they were assured of data confidentiality. When individuals were under 18 years old, written informed consent was obtained from a parent and/or legal guardian.

The study enrolled 51 patients, which were distributed into three groups, named: (i) orthodontic, (ii) TMD, and (iii) DFD. The orthodontic group included patients under orthodontic treatment, without skeletal deformities or any indication for orthognathic surgery. The TMD group was composed of patients with a TMD diagnosis, without either skeletal deformities or orthodontic treatment. Patients with skeletal deformities under orthodontic treatment, in preparation for orthognathic surgery, were included in the DFD group.

The participants were recruited from the Orthodontic, Occlusion, and Oral Surgery Units of the School of Health and Life Sciences (PUCRS, Brazil), from March to December 2021. Those patients with cleft lip/palate, deformity due to trauma or tumors, autoimmune diseases, systemic diseases, or those who previously underwent orthognathic or joint surgery were excluded from the study. Drinking alcohol and/or smoking were also considered exclusion criteria. The flowchart (Fig 1) summarizes the study design. The socio-demographic data, habits, type of deformity and main complaints of the participants are provided in Table 1.

The sample size was estimated based on the study by Vrbanovic et al. [14]. The authors evaluated the salivary levels of inflammatory biomarkers, oxidative stress, and cortisol in patients with TMD, correlating these findings with pain levels and psychologic factors, before and after the use of occlusion splint. The authors observed a significant difference in the levels of oxidative stress after the intervention, in a group of 12 patients. Considering a significance level of 5% and a power of 90%, an N = 17 patients/group was established (G*Power Version 3.1.9.7; Germany). The main outcomes were the results of both Axis I and Axis II of the DC/TMD. The secondary outcomes were the salivary levels of the biomarkers IL-1β, glutamate, and serotonin, besides the QoL data from the short version of the *Oral Health Impact Profile* (OHIP-14).

### DC-TMD evaluation

One trained examiner assessed all the individuals for TMD diagnosis by using the Brazilian Portuguese version of the DC/TMD [15]. This tool presents two axes: Axis I is related to temporomandibular joint (TMJ) physical signs and symptoms, whereas Axis II refers to psychological aspects. According to the Axis I (clinical examination form), it is possible to diagnose individuals based on a Diagnostic Decision Tree, with three groups of disorders, denoted: (i) myofascial pain; (ii) articular disorders; and (iii) cephalgia. The Axis II was measured by six questionnaires: Graded Chronic Pain Scale Version 2.0 (GCPS), Jaw Functional Limitations Scale-20-item (JFLS-20), PHQ-9 Depression, GAD-7 Anxiety, PHQ-15 Physical Symptoms, and Oral Behaviors Checklist (OBC). The GCPS is applied to investigate pain intensity, function, and the number of days with pain. This analysis can determine the Chronic Pain Grande (CPG) into five levels: none; low-intensity pain, without disability; high-intensity pain, without disability; moderately limiting; and severely limiting. The JFLS-20 is used to assess functional limitations of the masticatory system based on three subscales: masticatory limitation, vertical

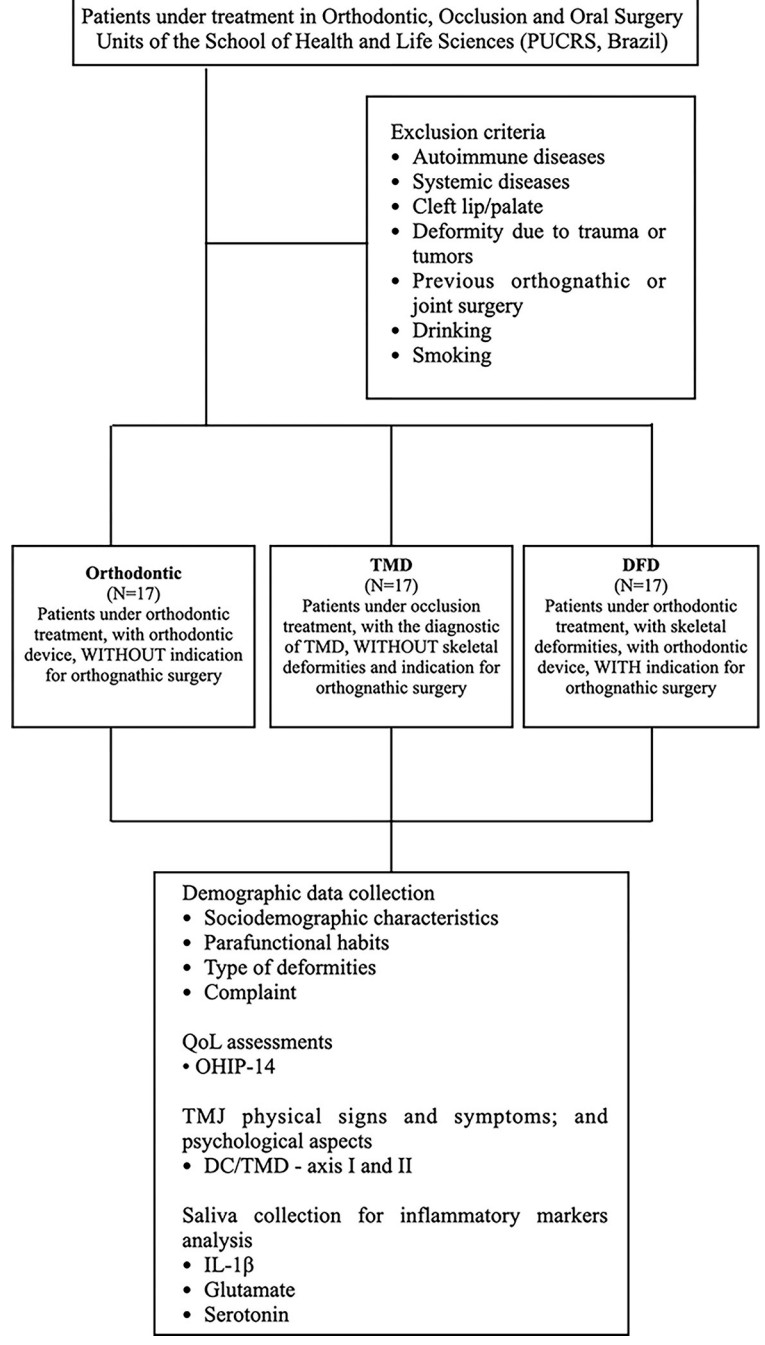

**Fig 1. Study design.** This is the flowchart of the Study Design.

mobility limitation, and verbal and non-verbal communication limitation, and it is comprised of a 20-item inventory. The PHQ-9 Depression is comprised of 9 items for assessing depressed mood, which could be interpreted into none; mild; moderate; moderately severe; and severe depression. The GAD-7 Anxiety includes seven items for assessing anxious mood and behavior. The anxiety items are scored as none; mild; moderate; and severe anxiety. The PHQ-15 is formed by 15 items for evaluating non-specific physical symptoms, also referred to as functional symptoms or medically unexplained symptoms. This scale can be interpreted as low;

**Table 1. Socio-demographic data of patients included in the study.**

| | Groups | | | |
| --- | --- | --- | --- | --- |
| | **Orthodontic** | **TMD** | **DFD** | ***p*-value** |
| **Parameters** | | | | |
| **N** | 17 | 17 | 17 | - |
| **Sex** | | | | |
| Male | 4 | 6 | 7 | 0.694 |
| Female | 13 | 11 | 10 | |
| **Age range (y)** | 16–51 | 18–66 | 19–47 | - |
| **Mean ± SD Age (y)** | 27.2 ± 8.025 | 36.5 ± 16.2* | 21.9 ± 7.8 | 0.001 |
| **Educational level** | | | | |
| Middle School | 2 | 3 | 1 | 0.026 |
| High School | 13 | 4 | 8 | |
| College | 2 | 10 | 8 | |
| **Ethnicity** | | | | |
| African American | 0 | 0 | 1 | 0.360 |
| Caucasian | 17 | 17 | 16 | |
| **Parafunctional habits** | | | | |
| Yes | 6 | 17 | 10 | 0.0003 |
| No | 11 | 0 | 7 | |
| **Type of deformity** | | | | |
| Class I | 17 | 17 | 0 | <0.0001 |
| Class II | 0 | 0 | 6 | |
| Class III | 0 | 0 | 11 | |
| **Complaint** | | | | |
| Aesthetics | 10 | 0 | 3 | <0.0001 |
| Functional | 7 | 17 | 6 | |
| Both | 0 | 0 | 8 | |

Abbreviations: DFD, dentofacial deformities; N, number of participants in each group; SD, standard deviation; TMD, temporomandibular disorder; y, years.

*Significant difference among the groups, as evaluated by Chi-Square test (frequency data) or one-way ANOVA followed by Tukey's post hoc test (for mean age).

medium; and high physical symptoms. The OBC is used to determine the presence of parafunctional behaviors, as a risk factor for TMD, and it is scored as none; low; and high risk.

## Assessment of QoL

The self-applied Brazilian Portuguese version of OHIP-14 was adopted to assess the patient's perception of oral health-related quality of life (OHRQoL) [16]. This instrument consists of 14 questions divided into seven domains: (i) functional limitation, (ii) physical pain, (iii) psychological discomfort, (iv) physical disability, (v) psychological disability, (vi) social disability, and (vii) handicap. Each domain is evaluated by two questions. The total points for the two corresponding answers provide the score for each domain, which ranges from 0 to 8. The total score is calculated as the sum of scores of all domains and it can vary from 0 to 56. Higher scores indicate poorer OHRQoL levels.

## Saliva collection

The methodology was based on the previous publication by Volkweis et al. [12]. Stimulated whole saliva was collected from 9 to 11 a.m. The patients had to be fasting for at least 1 h before

the sample collection. Following a mouth rinse with water, the patients were instructed to rest in a seated position and chew orthodontic rubber bands for 1 min. The saliva was collected for 5 min, in 5-ml sterile flasks and immediately stored at -20˚C until analysis.

## Determination of IL-1β levels

The salivary levels of interleukin-1β (IL-1β) were determined by the sandwich enzyme-linked immunosorbent assay (ELISA) using a DuoSet commercial kit (human IL-1β/IL-1F2; DY201-05), according to the fabricant instructions. The directions recommended saliva centrifugation and the use of the aqueous layer for the assay (R&D Systems; Minneapolis, MN, USA). The results were expressed in picograms per ml. The limits of detection were 3.91 to 250 pg/ml.

## Analysis of glutamate and serotonin levels

Salivary glutamate and serotonin were determined by liquid chromatography-tandem mass spectrometry (LC-MS/MS) preceded by a sample cleanup method based on protein precipitation by addition of organic solvent, as described elsewhere [17], with modifications. Briefly, frozen samples were thawed at room temperature, under reduced light conditions, and transferred (500 μl) to a 1.5 ml microtube. Two aliquots of 150 μl and one aliquot of 200 μl of acetonitrile were successfully added to each sample with 30 s of vortexing after each addition. The extracts were homogenized in an orbital mixer at 70 rpm for 10 min, and centrifuged at 14,000 rpm, at 4˚C, for 10 min. The supernatant was transferred to a plastic vial and injected into an LC-MS/MS system consisting of a Waters Xevo TQ-S Micro mass spectrometer equipped with an ESI source operated in positive mode and coupled to a UPLC Acquity 1 Class Plus liquid chromatography (Waters, Milford, MA, USA). Chromatographic separations were performed on a Zorbax SB-C18 (4.6 × 50 mm, 1.8 μm, Agilent Technologies, USA) using a mobile phase consisting of (A) 2 mM ammonium acetate and (B) methanol. The gradient started with a flow rate of 0.4 ml/min and 2% B, passing to 95% B from 1 to 2 min, and remaining in this condition for 2 min. After this period, the system was restored to the start condition to allow a new injection, after 1 min of equilibration. The total chromatographic run comprised 5 min, with the column temperature at 40˚C, and an injection volume of 5 μl. The spectrometer was operated in MRM mode to monitor Glutamate (m/z 148 > 84 for quantification; m/z 148 > 102 for confirmation) and serotonin (m/z 177 > 160 for quantification; m/z 177 > 115 for confirmation). Quantification was performed by external standardization and calibration curve at concentrations of 50 to 1500 ng/mL for glutamate, and 0.05 to 50.0 ng/mL for serotonin [17].

## Statistical analysis

Data are expressed as mean ± standard deviation (SD), except for the frequency results. Sociodemographic variables, questionnaire data, and biomarker levels were analyzed by the Chi-Square test or by one-way or two-way analysis of variance (ANOVA) followed by Tukey's post hoc test. Pearson's coefficients were used to correlate instrument-derived scores and biomarker levels. $P$-values less than 0.05 were considered indicative of significance. All the statistical tests and graphs were performed with GraphPad Software Prism 9.3.1 for Windows (GraphPad Software; San Diego, California, USA).

## Results

### Demographic data

The sample comprised 51 individuals, distributed into three groups, with 17 patients each. The age of participants ranged from 16 to 66. The mean age of the individuals in the orthodontic,

TMD, and DFD groups was 27.2 ± 8.0, 36.5 ± 16.2, and 21.9 ± 7.8 years, respectively. The percentage of females was superior in the three groups. Most of the participants had completed high school and 98% of the individuals were Caucasians. When considering parafunctional habits, all the patients included in the TMD group (n = 17) answered that they have any parafunctional habits. The most common type of deformity for the DFD group was Class III. The main complaint that led the patients to seek treatment in the orthodontic, TMD, and DFD groups was aesthetics, functional ailments, or both, respectively. This data set, accompanied by the respective *P* values, is summarized in Table 1.

### DC-TMD evaluation Axis I

The analysis of Axis I of DC-TMD (Fig 2) showed that both TMD and DFD groups presented higher frequencies for the three evaluated groups of disorders, namely myofascial pain, cephalgia, and articular disorders, in comparison with the orthodontic group (*p* = 0.013). Although

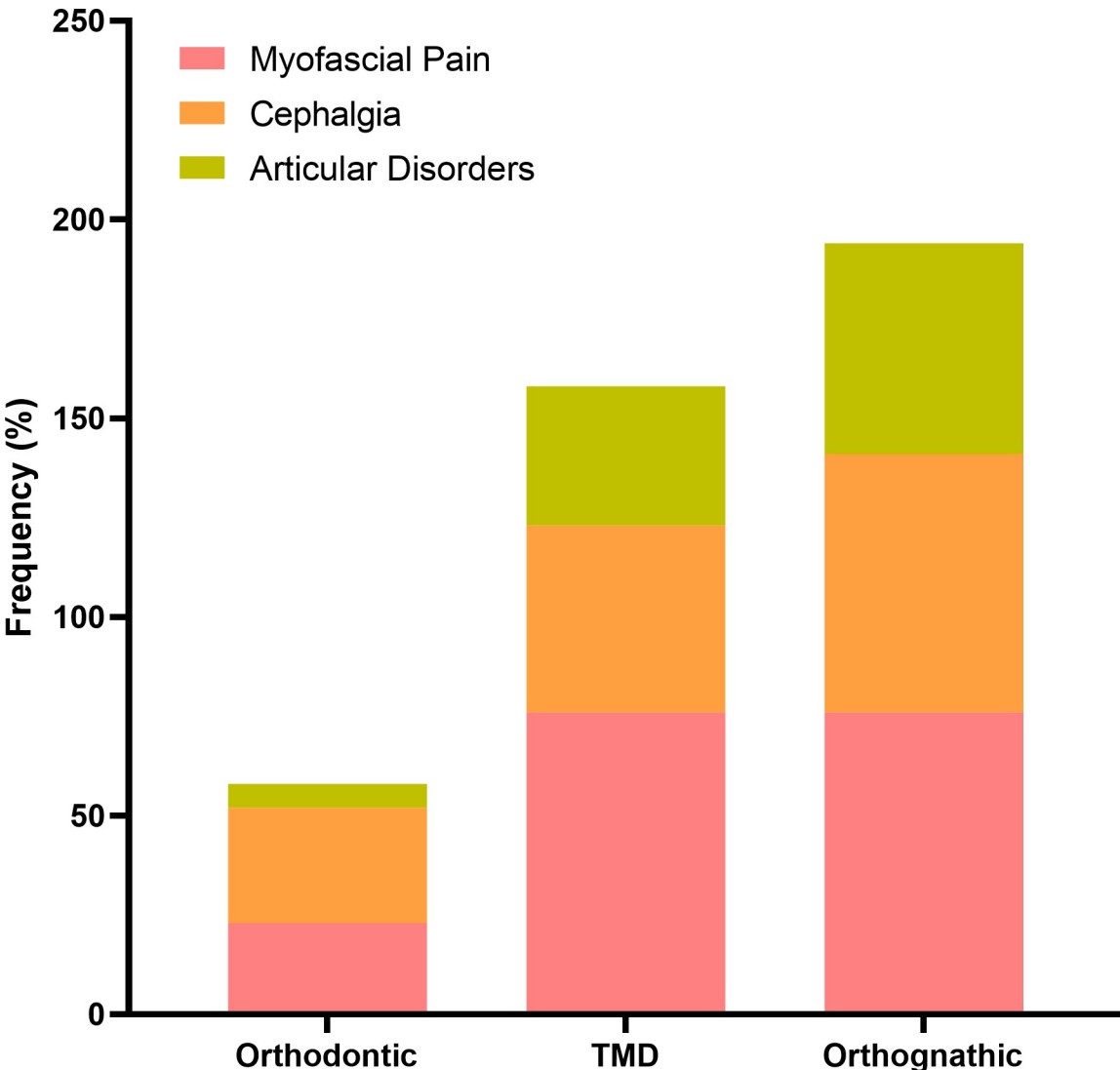

**Fig 2. Evaluation by Axis I of DC/TMD.** Frequencies (%) of myofascial pain, cephalgia, and articular disorders, as evaluated by DC-TMD Axis I for the orthodontic, TMD (temporomandibular disorder), and DFD (dentofacial deformity) groups.

no significant differences were observed when comparing only the TMD and the DFD groups ($p = 0.231$), there was a trend toward a higher frequency of individuals with cephalgia and articular disorders for the DFD group.

## DC-TMD evaluation Axis II

Regarding the pain components of DC-TMD Axis II, data depicted in Fig 3 demonstrates that individuals in the DFD group presented significantly higher levels of both characteristic pain intensity (panel A) and chronic pain grade (panel B), according to the evaluation by the GCPS. For the DFD participants, the levels of characteristic pain intensity were significantly greater when compared with the orthodontic ($p = 0.0009$) or the TMD ($p = 0.0067$) groups. Similarly, the chronic pain grade was significantly superior in DFD individuals, when compared with patients in the orthodontic group ($p < 0.0001$) or with a TMD diagnosis ($p = 0.003$).

For functional parameters of the Axis II, the OBC was significantly higher in either the TMD ($p = 0.0139$) or the DFD ($p < 0.0001$) groups, when compared with the orthodontic group. Moreover, the OBC scores were superior when comparing DFD vs. TMD patients ($p = 0.0106$) (Fig 4A). In the JFLS-20, the global analysis of masticatory functional limitations showed significantly higher scores in DFD vs. orthodontic ($p = 0.0146$) or TMD ($p = 0.0174$) groups. There were also significant differences when comparing the DFD group with orthodontic and TMD groups, regarding the subscales of masticatory quality ($p = 0.014$ and $p = 0.017$, respectively), and vertical mobility ($p = 0.0045$ and $p = 0.021$, respectively). Nonetheless, there were no significant differences when comparing the three groups concerning the subscale of verbal and non-verbal communication (Fig 4B).

From the analysis of the psychological components of Axis II, it was possible to observe that DFD individuals presented significantly higher scores for depression, anxiety, and non-specific physical symptoms compared to participants in the orthodontic or the TMD groups. For

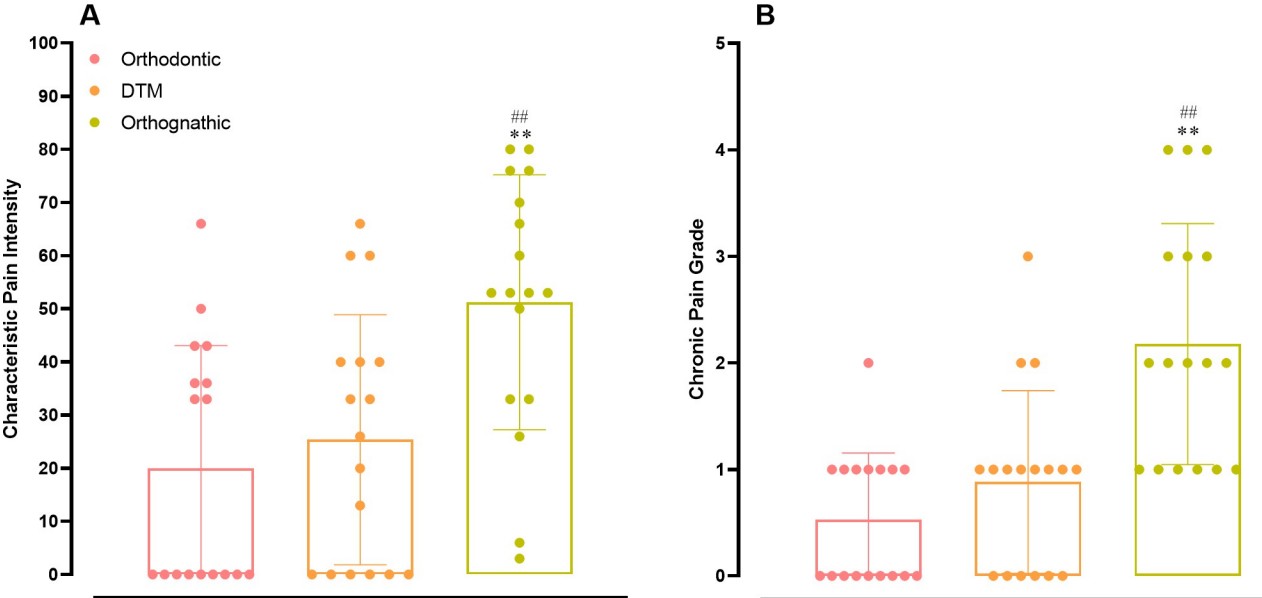

**Fig 3. Assessment of chronic pain in the different groups.** Evaluation of the components of CPGS (chronic pain grade scale), namely characteristic pain intensity (A) and chronic pain grade (B), for orthodontic, TMD, and DFD groups. Each column represents the mean, and the lines show the standard deviation (SD). The individual values are depicted as dot plots. **$p < 0.01$ DFD vs. orthodontic; ##$p < 0.01$ DFD vs. TMD. One-way ANOVA followed by Tukey's multi-comparison test. DFD, dentofacial deformity; TMD, temporomandibular disorder.

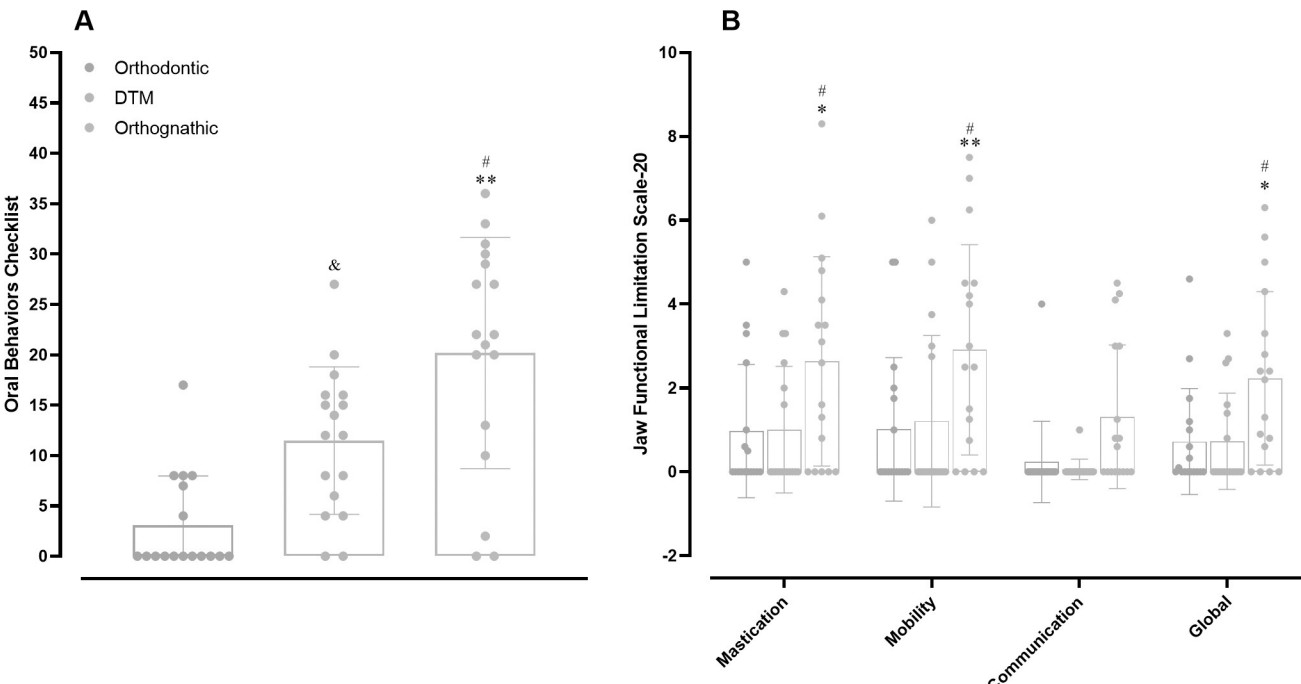

**Fig 4. Analysis of jaw functional habits and limitations.** (A) Comparison of oral behaviors checklist (OBC) assessment among orthodontic, TMD, and DFD participants. \*\*$p < 0.01$ DFD vs. orthodontic; #$p < 0.05$ DFD vs. TMD; &$p < 0.05$. One-way ANOVA followed by Tukey's multi-comparison test. DFD, dentofacial deformity; TMD, temporomandibular disorder. (B) Analysis of orthodontic, TMD, and DFD groups in the different subscales of the jaw functional limitation scale-20. \*$p < 0.05$; \*\*$p < 0.01$ DFD vs. orthodontic; #$p < 0.05$ DFD vs. TMD. Two-way ANOVA followed by Tukey's multi-comparison test. Each column represents the mean, and the lines show the standard deviation (SD). The individual values are depicted as dot plots. DFD, dentofacial deformity; TMD, temporomandibular disorder.

depression scores, as indicated by the PHQ-9 inventory, the comparison of DFD vs. orthodontic and TMD groups provided $p$ values of 0.0207 and 0.0004, respectively (Fig 5A). The GAD-7 anxiety questionnaire revealed significant differences when comparing DFD and orthodontic groups ($p = 0.0192$), or DFD vs. TMD individuals ($p = 0.0007$) (Fig 5B). Finally, an evaluation PHQ-15 scores demonstrated significantly superior scores for DFD participants vs. orthodontic ($p = 0.0004$) and TMD ($p = 0.0028$) groups (Fig 5C).

An overall analysis of the frequencies for each group in the PHQ-9 (panel A), GAD-7 (panel B), PHQ-15 (panel C), OBC (panel D), and chronic pain grade (panel E) instruments is provided in S1 Fig. In this data set, TMD and DFD groups presented high intensity of symptoms in the OBC evaluation. Also, moderate-to-severe limitations in the chronic pain grade assessment were observed only for the TMD and DFD participants. For the other instruments, individuals in the three groups were present independent of the magnitude of symptoms.

## Evaluation of QoL

The OHIP-14 instrument was used to evaluate the QoL of the participants. Data from global analysis demonstrated that DFD individuals displayed significantly higher scores for this questionnaire ($p < 0.0001$), in comparison with both orthodontic and TMD groups (Fig 6A). When each dimension of the OHIP-14 instrument was analyzed separately, it was possible to observe that DFD participants presented significantly higher scores for functional limitation, when compared with the TMD group ($p = 0.0395$). The scores were significantly higher for physical pain when comparing DFD vs. orthodontic groups ($p = 0.0139$). For the other five dimensions of OHIP-14, namely psychological discomfort, physical disability, psychological

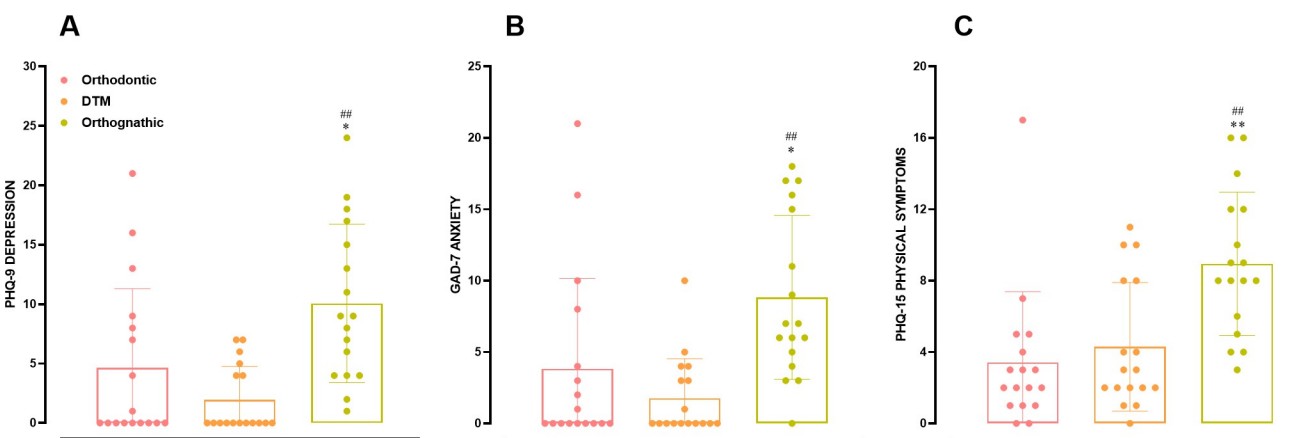

**Fig 5. Psychological parameters of DC/TMD Axis II.** Evaluation of PHQ-9 (A), GAD-7 (B), and PHQ-15 (C) in the orthodontic, TMD, and DFD groups. Each column represents the mean, and the lines show the standard deviation (SD). The individual values are depicted as dot plots. $*p < 0.05$; $**p < 0.01$ DFD vs. orthodontic; $\#\#p < 0.01$ DFD vs. TMD. One-way ANOVA followed by Tukey's multi-comparison test. DFD, dentofacial deformity; TMD, temporomandibular disorder.

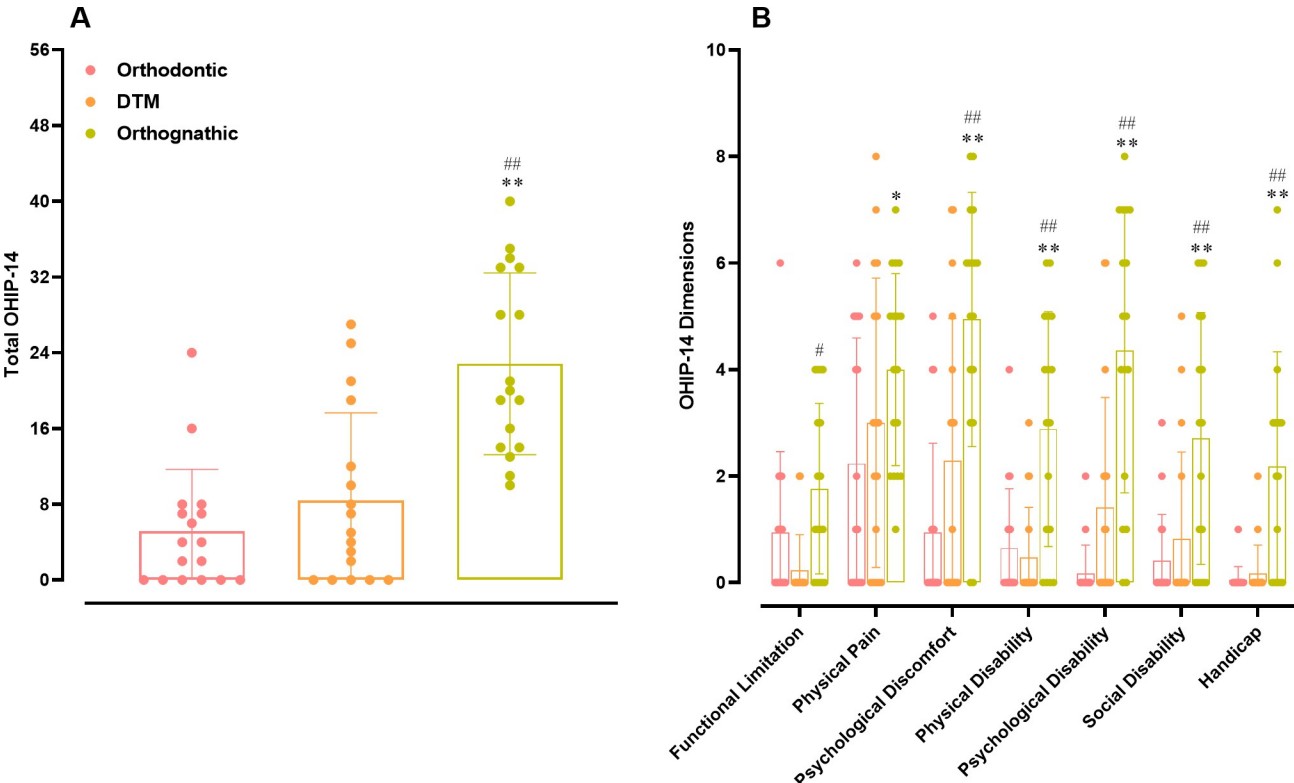

**Fig 6. Life quality assessment.** (A) Comparison of OHIP-14 instrument totals among orthodontic, TMD, and DFD participants. $**p < 0.01$ DFD vs. orthodontic; $\#\#p < 0.01$ DFD vs. TMD. One-way ANOVA followed by Tukey's multi-comparison test. DFD, dentofacial deformity; TMD, temporomandibular disorder. (B) Analysis of orthodontic, TMD, and DFD groups in the different domains of the OHIP-14 questionnaire. $*p < 0.05$; $**p < 0.01$ DFD vs. orthodontic; $\#p < 0.05$; $\#\#p < 0.01$ DFD vs. TMD. Two-way ANOVA followed by Tukey's multi-comparison test. Each column represents the mean, and the lines show the standard deviation (SD). The individual values are depicted as dot plots. DFD, dentofacial deformity; TMD, temporomandibular disorder.

disability, social disability, and handicap, the scores for DFD individuals were significantly higher, when compared with either the orthodontic or the TMD individuals ($p < 0.001$) (Fig 6B).

## Levels of salivary biomarkers

ELISA assay demonstrated that DFD individuals presented significantly elevated salivary levels of IL-1β, in comparison with patients in the orthodontic group ($p = 0.0062$), but not when compared with TMD patients ($p = 0.3660$). A comparison between orthodontic and TMD individuals indicated a trend toward higher IL-1β salivary contents in the latter group ($p = 0.074$) (Fig 7A). The salivary IL-1β amounts did not display any correlation with DC-TMD or QoL parameters (Table 2).

LC-MS evaluation demonstrated the presence of glutamate (Fig 7B) and serotonin (Fig 7C) in the saliva samples of all the analyzed groups, without significant differences among them, with $p$ values of 0.321 and 0.341, respectively. Strikingly, a significant positive correlation was detected for salivary glutamate levels and the DC-TMD Axis II parameters, namely CPG, OBC, and GAD-7. The salivary glutamate amounts also had a positive correlation with the total scores of the OHIP-14 QoL instrument. Besides, it was possible to observe a tendency for a positive correlation between salivary glutamate contents and the results of the PHQ-15 questionnaire. There was also a trend for a positive correlation between salivary levels of serotonin and the characteristic pain intensity from the GPCS. The correlation data is summarized in Table 2, showing the Pearson $r$ and the $p$ values.

## Discussion

This study compared the profile of patients with TMD diagnosis and DFD in preparation for orthognathic surgery, regarding both the Axis I and II of the DC-TMD instrument. A third group using an orthodontic device was included for comparison, as DFD participants are under pre-surgery orthodontic treatment. Our results demonstrate that TMD and DFD groups presented higher frequencies for the three evaluated groups of disorders, namely myofascial pain, cephalgia, and articular disorders, in comparison with the orthodontic group, according to the Axis I DC-TMD. Previous data indicated that 50–70% of all patients that present TMD refer to pain in the masticatory muscles [18]. Besides, it has been suggested that DFD might be

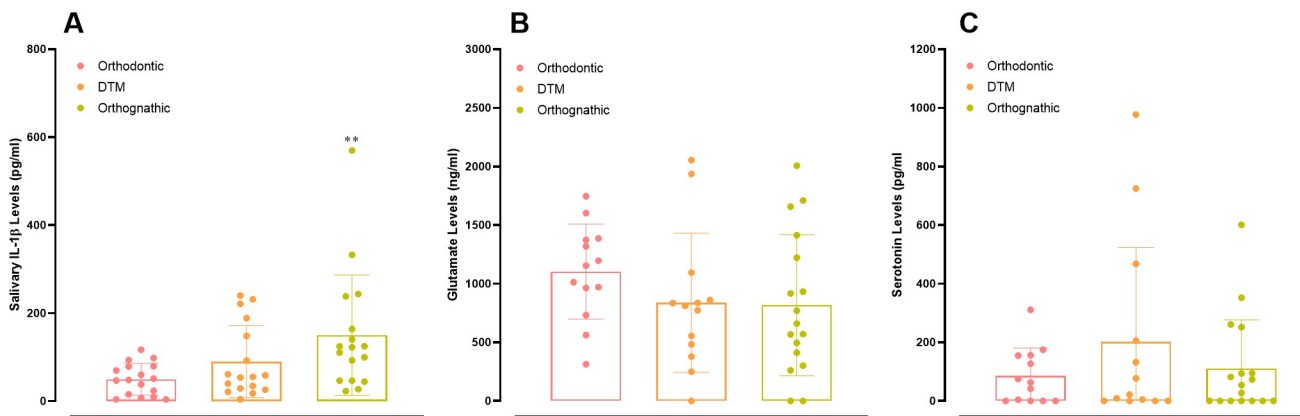

**Fig 7. Salivary biochemical markers.** Evaluation of salivary levels of IL-1β (A), glutamate (B), and serotonin (C) in the orthodontic, TMD, and DFD groups. Each column represents the mean, and the lines show the standard deviation (SD). The individual values are depicted as dot plots. **$p < 0.01$ DFD vs. orthodontic. One-way ANOVA followed by Tukey's multi-comparison test. DFD, dentofacial deformity; IL-1β, interleukin-1β; TMD, temporomandibular disorder.

**Table 2. Correlation between salivary biomarkers and DC-TMD results.**

| Salivary Biomarker | | | |
|---|---|---|---|
| **IL-1β** | | | |
| | **DC-TMD Parameter** | **Pearson r** | ***p-value*** |
| | CPI | 0.0584 | 0.350 |
| | CPG | 0.0270 | 0.528 |
| | OBC | 0.0381 | 0.452 |
| | JFLS-20 | 0.0275 | 0.524 |
| | PSQ-9 | 0.0002 | 0.950 |
| | GAD-7 | 0.0104 | 0.696 |
| | PSQ-15 | 0.0269 | 0.529 |
| Glutamate | | | |
| | **DC-TMD Parameter** | **Pearson r** | ***p-value*** |
| | CPI | 0.1043 | 0.206 |
| | CPG | 0.3630* | 0.010 |
| | OBC | 0.2417* | 0.045 |
| | JFLS-20 | 0.0460 | 0.408 |
| | PSQ-9 | 0.2467* | 0.042 |
| | GAD-7 | 0.1797‡ | 0.089 |
| | PSQ-15 | 0.3949** | 0.007 |
| 5-HT | | | |
| | **DC-TMD Parameter** | **Pearson r** | ***p-value*** |
| | CPI | 0.2271‡ | 0.053 |
| | CPG | 0.0239 | 0.553 |
| | OBC | 0.0744 | 0.289 |
| | JFLS-20 | 0.0768 | 0.281 |
| | PSQ-9 | 0.0515 | 0.380 |
| | GAD-7 | 0.0569 | 0.356 |
| | PSQ-15 | 0.0150 | 0.639 |

Abbreviations: CPI, characteristic pain intensity; CPG, chronic pain grade; GAD-7, general anxiety disorder-7; 5-HT, Serotonin; IL-1β, Interleukin-1β; JFLS-20, jaw functional limitation scale-20; OBC, oral behavior checklist; PHQ-9, patient health questionnaire-9; PHQ-15, patient health questionnaire-15. The Pearson's correlation coefficient was used to determine the correlation between each salivary biomarker and the DC-TMD variable. Significant correlation:

*$p < 0.05$

**$p < 0.01$.

‡ $p < 0.1 - > 0.05$.

a risk factor for TMD, probably because of the muscular and occlusal instability displayed by those patients [19]. A recent prospective cohort study carried out with 134 patients, using the Axis I of DC-TMD, verified that the prevalence of TMD in patients with mandibular asymmetry was significantly higher, suggesting that facial deformity might be a possible etiologic factor for TMD [20]. These findings support our data, considering that all the patients that presented DFD have been diagnosed with TMD.

The analysis of Axis II of the DC/TMD allows assessment of the severity of chronic pain and levels of depression and somatization in TMD patients [21]. The psychosocial assessments using the Axis II of DC-TMD are part of the contemporary biopsychosocial model for pain evaluation [18]. Although this instrument has not been used in previous studies with a similar

population like ours, some of the questionnaires composing the Axis II of DC-TMD have been used isolated for assessing the psychological aspects of DFD individuals, which have been considered in the present discussion.

OBC is a reliable questionnaire that quantifies the frequency of oral behaviors (OB) displayed during the month before the analysis. Our study showed that OBC was significantly higher in either TMD or DFD groups when compared with the orthodontic group. These results are somewhat in contrast with data published before [22], in which OBC did not differ between TMD and non-TMD groups. The authors proposed that OB should be divided into non-functional and functional activities, and had demonstrated that OBC subscale scores, representing non-functional activities, were higher in TMD groups when compared with the non-TMD group [22]. This notion is consistent with our data, as 17 out of 17 patients from the TMD group had parafunctional habits, a feature that has also been referred by 10 out of 17 patients from the DFD group. Previous studies suggest that OB might trigger overload and microtrauma to the masticatory system, being identified as a risk factor for developing painful TMD, in addition to muscular and occlusal instability [23,24]. It is important to mention that the present study is the first one to employ the OBC questionnaire for assessing DFD patients.

The muscle instability observed under skeletal malocclusion is likely associated with functional complaints in DFD patients. In the Axis II DC-TMD, the JFLS-20 instrument is employed to measure the functional limitations of the masticatory system. Herein, the patients in the DFD group showed higher scores for JFLS-20, regarding mastication and mobility outcomes, without significant changes in communication abilities. Our results were similar to other studies that found a higher prevalence of myofascial pain and functional limitations in patients that require orthognathic surgery [25,26]. It is possible to suggest that the JFLS-20 questionnaire might represent a useful tool for the analysis of functional aspects of individuals in preparation for orthognathic surgery.

It is recognized that individuals with DFD commonly present with psychological alterations, which are attributed to esthetics and functional problems, together with orofacial pain and chronic headaches [27]. Herein, the analysis of the psychological components of the Axis II of DC-TMD revealed higher scores of depression, anxiety, and non-specific physical symptoms in DFD participants, if compared with either orthodontic or TMD groups. Moreover, an analysis of GCPS, from Axis II DC-TMD, revealed higher painful symptoms in patients allocated in the DFD group. This instrument allows evaluation of the CPI and the CPG, as signs of chronic pain states. It has been demonstrated that individuals with DFD have five-fold higher chances of having depression, with a doubled risk of exhibiting chronic pain, allied with elevated scores of somatization and anxiety [26]. Moreover, recent studies have indicated that depression and anxiety might represent risk factors for TMD-related pain, while influencing muscular activities, disturbing coordination among masticatory muscles around the TMJ, with changes in mechanical properties, finally causing pain [28].

The influence of DFD on the patient's quality of life (QoL) has been widely debated in the literature. DFD might lead to lower self-esteem and psychological distress, thereby interfering with the QoL [8]. Our data demonstrated that DFD individuals presented significantly higher scores for the global analysis of the OHIP-14, as well as, on each separate domain, when compared with the other two groups, namely orthodontic and TMD. Accordingly, a recent systematic review and meta-analysis concluded that DFD patients had poorer QoL levels before orthognathic surgery, with a post-surgery improvement of aesthetic, functional, social, and psychological aspects of OHIP-14 [29]. Furthermore, we have previously demonstrated that pre-surgery DFD patients presented poorer QoL scores, according to the evaluation of the Orthognathic QoL Questionnaire (OQLQ) [12], supporting somewhat the present results. However, in the same study, there were no significant differences when comparing DFD and

orthodontic patients, while using the World Health Organization Quality of Life BREF instrument (WHOQOL-BREF) [12]. It is worth mentioning that both OHIP-14 and OQLQ, but not WHOQOL-BREF, are composed of questions addressing the dentofacial aspects of QoL.

We aimed to correlate the DC-TMD and QoL findings with the salivary levels of mediators that play a relevant role in pain, depression, and anxiety, specifically IL-1β, glutamate, and serotonin. Our results are in accordance with another recent study showing that DFD individuals present significantly higher salivary levels of IL- 1β when compared with patients under orthodontic treatment, but without DFD [12]. There was also a tendency for higher salivary IL-1β amounts in TMD patients, a finding that is supported by a previous study carried out with patients with TMD and fibromyalgia [5]. The elevated IL-1β salivary levels might well be related to the social context, as well as, to the oral function disability and pain observed in DFD individuals, which are seen to a lesser extent in TMD patients. This hypothesis is endorsed by prior data suggesting that DFD might be considered a continuous stressor modulating the expression of IL-6, which is influenced by a genetic polymorphism that contributes to TMD and depression, impacting the QoL [7].

We detected a significant positive correlation between glutamate contents and OHIP-14 QoL scores. A similar association has been established before, as salivary glutamate concentrations were positively correlated with the function domain scores of the OQLQ in DFD patients [12]. When analyzed together, previous, and present data allow suggesting that glutamate is likely implicated in the functional alterations seen in DFD patients. Notwithstanding, we also demonstrate that salivary glutamate levels showed a positive correlation with CPG, OBC, and PSQ-15 outcomes, with a mild correlation with the GAD-7 anxiety instrument. Therefore, the role of glutamate in DFD individuals might surpass the functional deficits, likely mediating chronic pain and somatization in those individuals. A recent study revealed high levels of salivary glutamate in patients diagnosed with TMD-related alterations, such as myalgia, which can be observed in our study as well [30]. For serotonin, there was a trend for a positive correlation between the salivary contents of this neurotransmitter and CPI. At the periphery, serotonin has been described as a pro-nociceptive mediator, contributing to chronic muscle pain, as observed in DFD individuals carrying out TMD [31]. In agreement, it was demonstrated that experimental tooth clenching, in TMD patients with myofascial pain, led to an increase in masseter serotonin levels [32]. This allows suggesting that muscle instability in DFD patients further contributes to painful symptoms, likely via the peripheral release of glutamate and serotonin, amplifying the psychosocial deficits in this population.

Regarding the main limitations of the present study, it is possible to indicate the small sample size, in addition to the fact that the patients have not been evaluated again after the orthognathic surgery, which remains to be investigated in future studies. Moreover, the sample consisted of individuals seeking for treatment in a university setting, not including patients recruited in clinical practice.

## Conclusion

Collectively, our data demonstrate that DFD patients have high scores for either Axis I or II of DC-TMD, which point out a TMD diagnosis. These individuals displayed functional limitations and detrimental OB, with intense chronic pain, allied to depression, anxiety, and nonspecific complaints. The alterations detected by the DC/TMD instrument are likely associated with low QoL outcomes and elevated salivary levels of the inflammatory cytokine IL-1β. Moreover, psychological, and painful complaints had a positive correlation with the release of the neurotransmitters in saliva, particularly glutamate, and to lesser extent serotonin. Our results

bring novel evidence on the impacts of TMD on patients carrying DFD, indicating the need of interdisciplinary approaches to improve life quality outcomes of those individuals.

## Supporting information

**S1 Fig. Frequencies of different parameters of Axis II DC/TMD.** Frequencies (%) for PHQ-9 (A), GAD-7 (B), PHQ-15 (C), Oral behavior checklist (D), and chronic pain grade (E), as evaluated by DC-TMD Axis II for the orthodontic, TMD (temporomandibular disorder), and DFD (dentofacial deformity) groups.
(PDF)

**S1 File. Raw data regarding the present manuscript.** Descriptive data and statistical analysis regarding the results presented in the manuscript.
(PDF)

## Author Contributions

**Conceptualization:** Betina B. Crescente, Maria M. Campos.

**Data curation:** Betina B. Crescente, Natalia V. Bisatto, Maria M. Campos.

**Formal analysis:** Betina B. Crescente, Maria M. Campos.

**Funding acquisition:** Betina B. Crescente, Maria M. Campos.

**Investigation:** Betina B. Crescente, Maria M. Campos.

**Methodology:** Betina B. Crescente, Maria M. Campos.

**Project administration:** Betina B. Crescente, Maria M. Campos.

**Resources:** Maria M. Campos.

**Software:** Maria M. Campos.

**Supervision:** Gabriel Rübensam, Guilherme G. Fritscher, Maria M. Campos.

**Validation:** Maria M. Campos.

**Visualization:** Maria M. Campos.

**Writing – original draft:** Betina B. Crescente, Maria M. Campos.

**Writing – review & editing:** Betina B. Crescente, Maria M. Campos.

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
