## [Decision Letter · Decision Letter 0]

29 May 2023

PONE-D-23-12302Screening for temporomandibular disorders in patients with dentofacial deformities: impacts on life quality and salivary biomarkersPLOS ONE

Dear Dr. Campos,

Thank you for submitting your manuscript to PLOS ONE. After careful consideration, we feel that it has merit but does not fully meet PLOS ONE’s publication criteria as it currently stands. Therefore, we invite you to submit a revised version of the manuscript that addresses the points raised during the review process.

We look forward to receiving your revised manuscript.

Kind regards,

Martina Ferrillo

Academic Editor

PLOS ONE

   "This work was supported by grants from Coordenação de Aperfeiçoamento de Nível Superior (CAPES; Financial Code 001); Conselho Nacional de Desenvolvimento Científico e Tecnológico (CNPq); and Fundação de Amparo à Pesquisa do Estado do Rio Grande do Sul (FAPERGS). B.B.C. is a MSc. Student 

receiving grants from CAPES. M.M.C. is a research career awardee of CNPq (304042/2018-8)"

   "BBC - (001) Coordenação de Aperfeiçoamento de Pessoal de Nível Superior 

NVB - (001) Coordenação de Aperfeiçoamento de Pessoal de Nível Superior 

MMC - (304042/2018-8) Conselho nacional de Desenvolvimento Científico e Tecnológico "

Additional Editor comments:

Please modify the paper according to the reviewers' suggestions. The paper will be reconsidered for publication.

Reviewers' comments:

Reviewer's Responses to Questions

**Comments to the Author**

1. Is the manuscript technically sound, and do the data support the conclusions?

Reviewer #1: Yes

Reviewer #2: Yes

2. Has the statistical analysis been performed appropriately and rigorously? 

Reviewer #1: Yes

Reviewer #2: Yes

3. Have the authors made all data underlying the findings in their manuscript fully available?

Reviewer #1: Yes

Reviewer #2: Yes

4. Is the manuscript presented in an intelligible fashion and written in standard English?

Reviewer #1: Yes

Reviewer #2: Yes

5. Review Comments to the Author

Reviewer #1: Dear Authors,

thank you for giving me the opportunity to revise your paper entitled " Screening for temporomandibular disorders in patients with dentofacial deformities: impacts on life quality and salivary biomarkers". The paper aims to explore the incidence of TMD in dentofacial deformities. The MS is well written and succinct, but there are some critical issues to ad-dress that makes the paper unsuitable for publication in Journal:

1. The Title is quite confuse and doesn’t explain the study sample. Please, change it

2. Please, delete in Abstract the sentence “in orthodontic treatment before surgical correction; n=17 each”

3. The introduction is quite poor of different clinical information. I suggest to improve it in terms of possible therapeutic approaches in TMD. Please, refer also to new conservative approaches ( “Marotta N, Ferrillo M, Demeco A, Drago Ferrante V, Inzitari MT, Pellegrino R, Pino I, Russo I, de Sire A, Ammendolia A. Effects of Radial Extracorporeal Shock Wave Therapy in Reducing Pain in Patients with Temporomandibular Disorders: A Pilot Randomized Controlled Trial. Applied Scien-ces. 2022”; 12(8):3821. https://doi.org/10.3390/app12083821)

5. Please, add the study design in the introduction

6. Please, delete in Material and Methods the sentence in brackets (n=17 each).

7. Results are well written

8. Please, add the clinical implication of your founding

9. Please, add the Conclusion

Best Regards

Reviewer #2: Dear Corresponding Author,

The paper is really interesting, well conducted and fits the objectives of the journal; but it is necessary to review some points in order to improve the quality of the paper:

1) First, i ask you to check the plagiarism of your article using Ithenticate

2) About the Title of the article, I suggest you to modify it and add the type of article.

3) The Abstract is precisely written, and the aim of the study is mentioned.

4) The introduction section is very short and is needed to add other references to increase the quality of the manuscript. I suggest you add some lines on the impact of quality of life in patients with TMD and the prevalence of TMD in children and adolescents. I suggest including the following articles in the bibliography to enhance the impact of the paper. [10.1111/joor.13446]; [10.1111/joor.13472]

5) The Material and Methods section is adequate and well organized, but punctuation and spaces between words should be reviewed.

6) The conclusion is in accordance with the objectives of the research, its results and their interpretation, as well as the relevant literature. I suggest you include a section on study limitation.

Regards

6. PLOS authors have the option to publish the peer review history of their article (what does this mean?). If published, this will include your full peer review and any attached files.

Reviewer #1: No

Reviewer #2: No

---

## [Author Response · Author response to Decision Letter 0]

15 Jun 2023

Reviewer #1:

Dear Authors,

thank you for giving me the opportunity to revise your paper entitled " Screening for temporomandibular disorders in patients with dentofacial deformities: impacts on life quality and salivary biomarkers". The paper aims to explore the incidence of TMD in dentofacial deformities. The MS is well written and succinct, but there are some critical issues to address that makes the paper unsuitable for publication in Journal.

Thank you very much for evaluating our paper. We have responded to your comments below and the paper has been revised accordingly. The corrections will certainly help to improve the scientific level of our manuscript. We hope that the revised version of the paper has been satisfactorily improved to permit the paper to be published in PLOS ONE. 

1. The Title is quite confuse and doesn’t explain the study sample. Please, change it

Thank very much for this comment. Please, observe that we have re-written the Title in order to address your concerns, also considering the comments of Reviewer 2. Thus, the paper is now entitled “Assessment of temporomandibular disorders and their relationship with life quality and salivary biomarkers in patients with dentofacial deformities: a clinical observational study”. 

2. Please, delete in Abstract the sentence “in orthodontic treatment before surgical correction; n=17 each”

Thank you very much for this comment. As recommended, we have removed the text in parenthesis. 

3. The introduction is quite poor of different clinical information. I suggest to improve it in terms of possible therapeutic approaches in TMD. Please, refer also to new conservative approaches 

(“Marotta N, Ferrillo M, Demeco A, Drago Ferrante V, Inzitari MT, Pellegrino R, Pino I, Russo I, de Sire A, Ammendolia A. Effects of Radial Extracorporeal Shock Wave Therapy in Reducing Pain in Patients with Temporomandibular Disorders: A Pilot Randomized Controlled Trial. Applied Sciences. 2022”; 12(8):3821. https://doi.org/10.3390/app12083821)

Thank you very much for this valuable criticism. Please, note that we have made an effort to improve the introduction section regarding the possible therapeutic approaches for TMD, also including the paper indicated by this Reviewer, in addition to another reference (Brighenti N, Battaglino A, Sinatti P, Abuín-Porras V, Sánchez Romero EA, Pedersini P, Villafañe JH. Effects of an Interdisciplinary Approach in the Management of Temporomandibular Disorders: A Scoping Review. Int J Environ Res Public Health. 2023 Feb 4;20(4):2777. doi: 10.3390/ijerph20042777). We hope this is now suitable to allow the publication of the manuscript in the present form. 

4. Please, add the study design in the introduction

As recommended, we have added the study design in the introduction section, in the last paragraph. Thank you for this criticism

5. Please, delete in Material and Methods the sentence in brackets (n=17 each).

As suggested, we have removed “(n-17 each)”.

6. Results are well written

Thank you very much for this positive comment. 

7. Please, add the clinical implication of your founding

As indicated, we have included a statement with the study clinical implications at the end of the conclusion. 

8. Please, add the Conclusion

As recommended, we have separated the last paragraph of the Discussion as the item “Conclusion”

Best Regards

Thank you very much for all the valuable comments regarding our manuscript. Your appointments certainly contributed to improve the quality of our study.  

Manuscript number PONE-D-23-12302

Response to reviewer 2

Reviewer #2:

Dear Corresponding Author,

The paper is really interesting, well conducted and fits the objectives of the journal; but it is necessary to review some points in order to improve the quality of the paper.

Thank you very much for revising our manuscript. Your comments will certainly contribute to improve the paper’s quality and readability. 

1) First, i ask you to check the plagiarism of your article using Ithenticate.

Thank you very much for your recommendation. Accordingly, we have checked the manuscript plagiarism by using the tool Turnitin, which is freely available in our Institution. Please, note that we have included the file with similarity analysis showing that parts of text that had been marked are mostly related to methodological aspects (such as questionnaire descriptions) or specific terms and author’s affiliation, without any characterization of plagiarism. The details regarding the maximal percentages of similarity are at the end of the PDF file. 

2) About the Title of the article, I suggest you to modify it and add the type of article.

Thank very much for this comment. Please, observe that we have re-written the Title in order to address your concerns, also considering the comments of Reviewer 1. Thus, the paper is now entitled “Assessment of temporomandibular disorders and their relationship with life quality and salivary biomarkers in patients with dentofacial deformities: a clinical observational study”. 

3) The Abstract is precisely written, and the aim of the study is mentioned.

Thank you very much for the positive comments about our abstract. Thank you again for the careful revision of our paper. 

4) The introduction section is very short and is needed to add other references to increase the quality of the manuscript. I suggest you add some lines on the impact of quality of life in patients with TMD and the prevalence of TMD in children and adolescents. I suggest including the following articles in the bibliography to enhance the impact of the paper. [10.1111/joor.13446]; [10.1111/joor.13472]

Thank you very much for your comments. Please, observe that we have extended the Introduction section of the study, by adding the abovementioned references, in addition to other two new references. 

5) The Material and Methods section is adequate and well organized, but punctuation and spaces between words should be reviewed.

Thank you very much for this comment and the positive feedback. As recommended this sections has been thoroughly revised regarding the spaces and punctuation. 

6) The conclusion is in accordance with the objectives of the research, its results and their interpretation, as well as the relevant literature. I suggest you include a section on study limitation.

Thank you very much. As recommended, we have included a separated paragraph with the main study limitations. 

Regards

We would like to thank this Reviewer again for the suggestions and criticisms regarding our manuscript.

---

## [Decision Letter · Decision Letter 1]

7 Jul 2023

Assessment of temporomandibular disorders and their relationship with life quality and salivary biomarkers in patients with dentofacial deformities: a clinical observational study

PONE-D-23-12302R1

Dear Dr. Maria M. Campos,

We’re pleased to inform you that your manuscript has been judged scientifically suitable for publication and will be formally accepted for publication once it meets all outstanding technical requirements.

Kind regards,

Martina Ferrillo

Academic Editor

PLOS ONE

Additional Editor Comments (optional):

Authors modified the text according to the suggestions.

I found this work impactful and it fits well with in the scope of this journal.

In my opinion, it is suitable for publication.

Reviewers' comments:

Reviewer's Responses to Questions

**Comments to the Author**

1. If the authors have adequately addressed your comments raised in a previous round of review and you feel that this manuscript is now acceptable for publication, you may indicate that here to bypass the “Comments to the Author” section, enter your conflict of interest statement in the “Confidential to Editor” section, and submit your "Accept" recommendation.

Reviewer #1: All comments have been addressed

Reviewer #2: All comments have been addressed

2. Is the manuscript technically sound, and do the data support the conclusions?

Reviewer #1: Yes

Reviewer #2: Yes

3. Has the statistical analysis been performed appropriately and rigorously? 

Reviewer #1: Yes

Reviewer #2: Yes

4. Have the authors made all data underlying the findings in their manuscript fully available?

Reviewer #1: Yes

Reviewer #2: Yes

5. Is the manuscript presented in an intelligible fashion and written in standard English?

Reviewer #1: Yes

Reviewer #2: Yes

6. Review Comments to the Author

Reviewer #1: (No Response)

Reviewer #2: Dear authors,

I am satisfied with the changes made

The article is well carried out and the changes have been incorporated and implemented in the text

Regards

7. PLOS authors have the option to publish the peer review history of their article (what does this mean?). If published, this will include your full peer review and any attached files.

Reviewer #1: **Yes: **Dario calafiore

Reviewer #2: No

---

## [Editor Report · Acceptance letter]

12 Jul 2023

PONE-D-23-12302R1 

Assessment of temporomandibular disorders and their relationship with life quality and salivary biomarkers in patients with dentofacial deformities: a clinical observational study 

Dear Dr. Campos:

I'm pleased to inform you that your manuscript has been deemed suitable for publication in PLOS ONE. Congratulations! Your manuscript is now with our production department. 

Kind regards, 

on behalf of

Dr. Martina Ferrillo 

Academic Editor

PLOS ONE